# The Golgi Apparatus: A Key Player in Innate Immunity

**DOI:** 10.3390/ijms25074120

**Published:** 2024-04-08

**Authors:** Ion Mărunţelu, Alexandra-Elena Constantinescu, Razvan-Adrian Covache-Busuioc, Ileana Constantinescu

**Affiliations:** 1Immunology and Transplant Immunology, Carol Davila University of Medicine and Pharmacy, 020021 Bucharest, Romania; 2Centre of Immunogenetics and Virology, Fundeni Clinical Institute, 022328 Bucharest, Romania; 3Faculty of Medicine, Carol Davila University of Medicine and Pharmacy, 020021 Bucharest, Romania; alexandra-elena.constantinescu0720@stud.umfcd.ro (A.-E.C.); razvan-adrian.covache-busuioc0720@stud.umfcd.ro (R.-A.C.-B.); 4“Emil Palade” Center of Excellence for Young Researchers (EP-CEYR), Romanian Academy of Scientists (AOSR), 050094 Bucharest, Romania; 5Romanian Academy of Scientists (AOSR), 050094 Bucharest, Romania

**Keywords:** immunology, innate immunity, Golgi-apparatus, intracellular-signaling, NLRP3-inflammasome

## Abstract

The Golgi apparatus, long recognized for its roles in protein processing and vesicular trafficking, has recently been identified as a crucial contributor to innate immune signaling pathways. This review discusses our expanding understanding of the Golgi apparatus’s involvement in initiating and activating these pathways. It highlights the significance of membrane connections between the Golgi and other organelles, such as the endoplasmic reticulum, mitochondria, endosomes, and autophagosomes. These connections are vital for the efficient transmission of innate immune signals and the activation of effector responses. Furthermore, the article delves into the Golgi apparatus’s roles in key immune pathways, including the inflammasome-mediated activation of caspase-1, the *cGAS*-STING pathway, and TLR/RLR signaling. Overall, this review aims to provide insights into the multifunctional nature of the Golgi apparatus and its impact on innate immunity.

## 1. Introduction

Camillo Golgi’s 1898 discovery of the Golgi apparatus using silver nitrate staining marked a pivotal moment. Initially called the internal reticular apparatus, it was later recognized as a crucial cellular component rather than an artifact [1].

The Golgi apparatus is vital for modifying and sorting newly synthesized proteins and lipids. Its dynamic structure, composed of stacked cisternae layers, undergoes regulated disassembly and reassembly during cell cycles. Positioned near nuclei in mammalian cells, it responds to stressors and mediates the transport of proteins and lipids from the endoplasmic reticulum (ER) to the trans-Golgi network (TGN) [2,3].

Eukaryotic cells rely on the secretory pathway for protein synthesis, processing, and delivery. Proteins originating in the endoplasmic reticulum progress to the Golgi complex for further processing and sorting. The Golgi’s role in cargo movement is debated between the cisternal maturation and vesicular transport models. The Cisternal Maturation Model envisions cisternae maturation from the ER to the TGN cisterna, while the Vesicular Transport Model emphasizes compartmental stability and vesicular trafficking [4].

The Golgi relies on various molecules involved in the trafficking process, such as tethers, SNAPs (soluble N-ethylmaleimide-sensitive factor attachment proteins), and SNAREs (SNAP receptors) to function properly [5,6]. Disruptions in the functionalities of these molecules at the level of the Golgi apparatus due to pathological conditions like exposure to monocrotaline pyrrole and nitric oxide scavenging can lead to significant alterations in the Golgi’s normal functioning and trafficking processes [7]. These alterations, in turn, affect the density and functionality of plasma membrane receptors critical for immune functions, contributing to the pathogenesis and progression of diseases such as pulmonary arterial hypertension [8,9,10,11].

Beyond its traditional functions, the Golgi participates in the initiation and activation of the innate immune response. Innate immunity represents the body’s first line of defense against invading pathogens and cancer. Unlike the adaptive immune system, which targets specific pathogens, the innate immune system provides a non-specific response and is ancient in evolutionary terms. It includes physical barriers like the skin and mucosa, as well as cellular and molecular components such as phagocytes, inflammatory serum proteins, and cytokines. Dendritic cells, macrophages, neutrophiles, eosinophiles and the complement system are the main cellular components of the innate immune system [12]. Innate immunity is mediated by a variety of pattern-recognition receptors (PRRs) such as Toll-like receptors (TLRs), RIG-I-like receptors (RLRs), Nod-like receptors (NLRs), AIM2-like receptors (ALRs), C-type lectin receptors (CLRs), and other DNA sensors [13,14]. PRRs detect microbial invasions (pathogen-associated molecular patterns/PAMPs) or danger signals (damage-associated molecular patterns/ DAMPs) [12,15]. Certain microbial stimuli and endogenous ligands can provoke durable changes in the function of innate immune cells, leading to increased responsiveness upon secondary stimulation (termed “trained immunity”) [16]. The differential regulation of innate immune memory across life stages helps build immunity in young individuals but may contribute to chronic inflammation in the elderly [17,18].

The controlled release of cytokines and inflammatory mediators in response to stimuli involves transcription, translation, post-translational modifications, and secretion via the ER, Golgi complex, or cell surface [19]. Cytokines such as IL-2, IL-3, IL-6, IL-10, IL-12, and TNF are directed to the ER for folding via signal peptides [19]. Properly folded cytokines are packaged into vesicles coated with coat protein complex II (COPII) for transport to the ER-Golgi intermediate compartment (ERGIC) and further through the medial-Golgi to the TGN for sorting and modifications [19]. Additionally, the Golgi apparatus plays a crucial role in cargo sorting, separating transmembrane and soluble cytokines, and contributing to protein trafficking [20,21]. The specificity and selectivity of cytokine secretion are controlled through various mechanisms involving SNAREs (soluble N-ethylmaleimide-sensitive factor attachment protein/SNAP receptors), which mediate the fusion of vesicles with the plasma membrane, and Rab GTPases, which regulate vesicle trafficking [19,22]. The expression and activity of these proteins can be cell-type specific, allowing for precise control over cytokine release [23,24,25]. Cytokine release can also be polarized, directing cytokines toward specific areas of the cell membrane to influence localized immune responses. For instance, during the formation of the immunological synapse between T cells and antigen-presenting cells, cytokines can be directed to this region to promote effective cell–cell communication and targeted immune responses [26]. Cytokines are released in the bloodstream by different cells, such as Th (T helper lymphocytes) and the innate immune system cells [27].

Different studies highlight trans-Golgi network (TGN) disruption during NLRP3 inflammasome activation, facilitating inflammasome assembly [28,29]. Inflammasomes are protein complexes that activate proinflammatory cytokines, such as IL-1b, through caspase-1-dependent pathways. The NLRP3 inflammasome’s distinct responsiveness to various stimuli, including antibiotics and ATP, has implications for various disorders.

STING (Stimulator of Interferon Genes), an ER-associated membrane protein, is crucial for pathogen recognition and activates type I interferon (IFN-I) pathways via the TBK1/IRF3 (TANK-binding kinase 1)/(interferon regulatory factor 3) signaling axis [30,31].

The innate immune system orchestrates inflammatory responses to infections and tissue damage. Pattern recognition receptors (PRRs), including Toll-like receptors (TLRs) and RIG-I-like receptors (RLRs), detect microbial and damage-related molecules. Non-professional immune cells, such as epithelial and endothelial cells, also participate in innate immune responses [32].

Toll-like receptor (TLR) transport to endosomes is meticulously regulated. TLR7′s interaction with Unc93B1 in the Golgi complex facilitates direct transport to endosomes, bypassing intermediate cell surface steps [33,34].

This review delves into the critical role of the Golgi apparatus in innate immunity, with a particular focus on its involvement in essential immune pathways, including inflammasome-mediated activation of caspase-1, *cGAS*-STING, and TLR/RLR signaling. It further explores the alterations in Golgi structure and function following microbial infections, elucidating how these changes impact the immune responses. The article aims to deepen understanding of these complex biological processes, the Golgi apparatus’s adaptive responses to microbial challenges, and its potential implications in treating inflammatory disorders.

## 2. Golgi Apparatus and NLRP3 Inflammasome

The Golgi apparatus has long been known for its role in the modification and transport of proteins. However, different studies have revealed the influence of the Golgi apparatus on the activation of inflammasomes (see Figure 1). Inflammasomes, complex molecular structures that are formed in response to pathogens or danger signals, are crucial components of the body’s defense mechanism. Their precise control is essential for effective host defense in complex organisms.

NLRP3 inflammasome localizes near the Golgi—an aspect that plays a pivotal role in inflammasome activation. Upon activation, NLRP3 directly interacts with mitochondria-associated endoplasmic reticulum membranes (MAMs), and substances capable of activating NLRP3 facilitate the proximity of MAMs to Golgi membranes [35]. Furthermore, Zhang et al. recently proved an increased level of diacylglycerol (DAG) in Golgi organelles [35]. When activated with inflammasome activators, MAMs localize near Golgi membranes; an increase in diacylglycerol DAG levels at Golgi promotes recruitment and activation of protein kinase D (PKD). PKD is a family of serine/threonine-protein kinases comprising PKD1, PKD2, and PKD3. They play a critical role in the export of TGN and maintaining the balance of lipids in Golgi. Abnormal activity of these kinases could be linked to cancer progression by potentially altering membrane trafficking [36,37,38]. Once activated, PKD phosphorylates NLRP3, leading to its release from MAMs and full maturation within the cytosol [35,39]. This spatial relationship between the Golgi apparatus and NLRP3 hints at the Golgi’s regulatory function in the assembly and activation of the NLRP3 inflammasome (see Figure 1). The NLRP3 inflammasome comprises three major components: the sensor protein NLRP3, the adaptor protein ASC (also known as PYCARD), and the effector protein caspase 1. NLRP3 belongs to the NOD-like receptor (NLR) protein family, consisting of proteins with an NOD-like domain (NACHT), an N-terminal effector domain such as Pyrin domains (PYDs), caspase recruitment domains (CARD) or baculovirus inhibitor of apoptosis protein repeat (BIR), and a C-terminal leucine-rich repeats (LRRs). An ATPase activity within their central NACHT domain facilitates receptor oligomerization while their N-terminal PYD or CARD interact with ASC/caspase-1 domains via interaction ATPase activity [40].

The activation of the NLRP3 Inflammasome involves two distinct signals: priming signals and activation signals. Priming signals are typically initiated by microbes or endogenous cytokines, while activation signals are triggered by pathogen-associated molecular patterns (PAMPs) or danger-associated molecular patterns (DAMPs). These signals set off a cascade of events, culminating in the assembly of the NLRP3 inflammasome [41].

Activation of the NLRP3 inflammasome triggers the maturation and release of proinflammatory cytokines such as IL-1β and IL-18 [42,43], crucial for initiating early inflammatory responses and pyroptosis [35]. The Golgi apparatus plays a pioneering role in these processes. This dynamic organelle, which is responsible for the modification and transport of proteins, contributes significantly to the assembly and activation of the NLRP3 inflammasome [44]. Signals like potassium (K^+^) or chloride ions (Cl^−^) eflux, calcium ion influx, disruption of lysosomes, mitochondrial dysfunction, metabolic changes, and metabolic disorders [44] may also be involved in NLRP3 activation.

Inflammasome activation plays an essential role in protecting against invading pathogens. Still, excessive activation and gain-of-function mutations of these proteins have been implicated in numerous inflammatory diseases, including gout, type 2 diabetes, Alzheimer’s disease, atherosclerosis, and arthritis—in addition to autoinflammatory conditions like CAPS/FMF/MAS syndromes and cryopyrin-associated periodic syndromes (CAPS/FMF/MAS) [37,45]. NLRP3 inflammasome has also been implicated in an inflammatory response triggered by dying cells without evidence of infection, which may contribute to tissue damage and worsen diseases such as acute kidney injury [46,47].

The involvement of the Golgi apparatus in inflammasome activation goes beyond its conventional tasks. As a central hub for cellular transport and modification processes, the Golgi apparatus plays a regulatory role in coordinating the precise positioning of components and molecular interactions [48]. This orchestration optimizes the assembly and activation of the NLRP3 inflammasome, contributing to an efficient and controlled response to viral infections. Taken together, these discoveries underscore the profound role of the Golgi apparatus as a central orchestrator of immune signaling events (see Figure 1). Its remarkable ability to regulate inflammasome activation, amplify immune responses, and intersect with intricate cellular processes, including cholesterol metabolism, underscores its importance.

The extensive utilization of the multiple contributions of the Golgi apparatus holds great promise for the development of innovative therapeutic strategies. These strategies, aiming to modulate immune responses, have the potential to alleviate inflammatory disorders and usher in a new era of targeted interventions.

In summary, the Golgi apparatus’s multifaceted contributions to inflammasome activation are increasingly recognized as essential for immune responses. Its role in modulating NLRP3 assembly, optimizing spatial relationships, and enhancing immune signaling sheds light on new dimensions of innate immunity regulation. Understanding the Golgi apparatus’s involvement holds the potential to shape novel therapeutic strategies, targeting immune responses and offering insights into the treatment of inflammatory disorders.

## 3. Golgi Apparatus and *cGAS*–STING Signaling

This section deals comprehensively with the complex interaction between the STING (Stimulator of Interferon Genes) molecule, the endoplasmic reticulum, and the Golgi apparatus. The STING adaptor protein plays a central role in innate immune responses. It activates both the NF-kB and IRF3 transcriptional pathways, leading to the production of type I interferons that promote a fundamental antiviral defense mechanism [49]. Furthermore, it has been shown that STING is also involved in cellular senescence, death, and autophagy [50], emphasizing the Golgi apparatus’s contribution to regulating these processes [49].

In healthy eukaryotic cells, DNA normally resides in the nucleus and mitochondria. The presence of self-DNA in the cytoplasm signals genomic DNA damage that occurs in cancer and many other conditions, including autoimmune and neurodegenerative diseases [51,52]. Non-self cytosolic DNA can be found in case of microbial infections such as simplex virus, vaccinia virus, cytomegalovirus, *Chlamydia trachomatis*, *Mycobacterium tuberculosis*, and *Francisella novicida* [53]. In each case, DNA serves as DAMP (damage-associated molecular pattern) or PAMP (pathogen-associated molecular pattern), which triggers innate immune mechanisms via PRR recognition and subsequently via the activation of the *cGAS*-*cGAMP*-STING signaling pathway.

The *cGAS*-STING pathway consists of two major components: the *cGAS* protein and STING. Cyclic guanosine monophosphate (GMP)- adenosine monophosphate (AMP) synthetase (*cGAS*) belongs to the nucleotidyl-transferase family and is a protein consisting of different regions necessary to create the catalytic pocket for *cGAS* substrates, GTP and ATP, and carry out cyclization reactions [54].

*cGAS* is present in the nucleus, where its tight tethering to chromatin prevents self-DNA from activating it [55]. *cGAS* detects cytosolic dsDNA in the plasma membrane [56]. *cGAS* binds to double-stranded DNA (dsDNA), resulting in the formation of *cGAS*-dsDNA complexes [57,58]. This interaction causes a structural change in the catalytic site of *cGAS*, activating its enzymatic activity to produce 2’3’-*cGAMP* from ATP and GTP [51]. The *cGAS*-DNA interactions lead to high-order oligomerization and the formation of liquid–liquid phase separation (LLPS), which is important in vivo for triggering the *cGAS*-STING signaling pathway [59]. However, studies have shown that mutations and truncations of *cGAS* and short DNA (<45 bp) can reduce or even eliminate *cGAS* activity by attenuating *cGAS*-DNA oligomerization and LLPS [59,60]. This underscores the importance of DNA length, *cGAS* protein integrity and the interaction valency between *cGAS* and the DNA in effectively activating the *cGAS*-STING pathway. Additionally, free zinc ions further boost *cGAS* enzyme activity [59]. Pathogen proteins can regulate the phase separation of *cGAS* as well. Streptavidin, secreted by the bacterium Streptomyces avidinii, binds to *cGAS* to enhance *cGAS*–DNA interactions and promote LLPS of this complex, leading to enhanced *cGAS* activation and interferon-β production [61]. Conversely, herpesvirus proteins *ORF52* and *VP22* disrupt *cGAS*-DNA droplet formation, inhibiting *cGAS* activity by forming new droplets composed of viral proteins and DNA [61]. Moreover, the phase separation limits the access of negative regulators, such as three-prime repair exonuclease 1 (TREX1) and barrier-to-autointegration factor (BAF), enhancing the cytosolic DNA sensing mechanism [62].

Unlike other cyclic dinucleotides found in bacteria, archaea, and protozoa, *cGAMP* contains a unique phosphodiester bond between the 2’ OH of GMP and the 5’ phosphate of AMP [63]. This unique phosphodiester bond is less susceptible to degradation by 3′-5’ phosphodiesterases, allowing *cGAMP* to bind to multiple allelic variants of STING found in the human population [64].

The calcium sensor, Stromal Interaction Molecule 1 (STIM1), regulates the activity and stability of STING within the endoplasmic reticulum (ER) before it binds to *cGAMP* (SEL1L–HRD1 endoplasmic reticulum-associated degradation controls STING-mediated innate immunity by limiting the size of the activable STING pool) [65]. STIM1 is vital in maintaining STING in its resting state within the ER [66]. However, upon encountering *cGAMP*, STING undergoes a translocation from the ER to the ER-Golgi intermediate compartment (ERGIC) and subsequently to the Golgi apparatus. This journey is not merely a passive relocation but is essential for the activation of STING [67]. The Golgi apparatus plays a crucial role in this process by facilitating the necessary modifications to STING that enable its full activation [68].

After its activation, STING translocates to the Golgi via COP-II carrier vesicles [69]. It has also been shown that the COP-I complex, which facilitates retrograde transport from the Golgi to the ER, has a major impact in modulating the *cGAS*-*cGAMP*-STING signaling pathway by ensuring STING retrieval to the ER and, therefore, its activation [50]. The collaboration between ERGIC and STING underscores the importance of Golgi-mediated transport in controlling the immune response. Furthermore, at the Golgi, STING recruits TBK-1 (TANK-binding kinase 1) and binds to it.

It is known that in order to bind to TBK-1, c-GAMP-bound STING molecules cluster in either tetramers or dimers, but we still do not know how STING polymerization occurs. Different studies have shown multiple models of possible activation mechanisms. Taguchi et al. [50] hypothesized that the palmitoylation of STING in the Golgi is essential for efficient signal activation, as it permits this protein to enter the lipid rafts that form at the TGN and recruit TBK-1. Treatment of cells with the palmitoylation inhibitor 2-bromopalmitate (2-BP), C-176, and C-178 (two nitrofuran derivatives) inhibits the type I interferon response by impairing STING palmitoylation at the Golgi and the formation of STING clusters, but without affecting its translocation from the ER [70]. Another study suggested that sulfated glycosaminoglycans (sGAGs) synthesized in the Golgi apparatus interact with STING to initiate its oligomerization [71]. STING binds sGAGs through its polar residues, and the strength of this electrostatic interaction determines the degree of STING activation. Thus, sGAGs were identified as necessary coligands for STING activation and downstream IFN signaling [71]. In the context of these intricate processes, the strategic role of the Golgi apparatus in facilitating these reactions comes to the fore, highlighting its contribution to the regulation of the immune response.

The Golgi apparatus facilitates the interaction of STING with TBK-1 and is not merely a passive waypoint. It ensures the proper phosphorylation of the C-terminal tail region of STING, which is crucial for creating a binding site for IRF3 (interferon regulatory factor 3) [72]. Once activated, IRF3 initiates a cascade of immunological events. IRF3 translocates to the nucleus and initiates the transcription of several immune response genes. Foremost among these is the gene for type I interferon-beta (IFNb), a critical cytokine in antiviral defense [49] and IFN alpha. TBK-1, in association with STING, also activates another transcription factor via phosphorylation, NF-kappaB (nuclear factor kappa B), increasing the production of proinflammatory cytokines such as IL-6 and TNF [73] and further amplifying the innate immune response to ds-DNA.

The cycle of STING culminates with its degradation, a process essential for the termination of the immune response. This degradation occurs in the lysosomes and is mediated by the endosomal sorting complexes required for transport (ESCRT)-driven microautophagy, as described by Kuchitsu et al. [74]. This lysosomal degradation represents a crucial regulatory checkpoint, ensuring that the immune response does not perpetuate indefinitely [74]. The orchestrated journey of STING from the ER, through the Golgi, to the lysosomes underscores the complex regulation of immune signaling pathways, with each organelle playing a specific and essential role in the modulation of the body’s defense mechanisms (Figure 2).

Regulatory mechanisms are needed to prevent the self-activation of *cGAS*/STING during mitosis. Initially, scientists thought that the cytosolic compartmentalization of *cGAS* could serve as such a mechanism [75]. However, more sophisticated strategies are needed to attenuate *cGAS*/STING activity during cell division, as mixing nuclear chromosomes and cytosolic compartments during mitotic nuclear envelope breakdown (NEBD) occurs. The chromatinized nature of genomic DNA present inside cells can act as a mitigating factor for *cGAS*/STING activity. Histones structurally mark DNA as “self”, which leads to the tight tethering of chromosome-bound *cGAS* to chromatin through interactions with H2a/H2b dimers [75]. This chromatin interaction results in weak activation of *cGAS* and low production of *cGAMP* [75]. During NEBD, *cGAS* localizes to condensed chromosomes. In the process of open mitosis, the response of *cGAS*/STING to transfected DNA decreases, and this is linked to the vesiculated state of the mitotic Golgi [75]. These findings suggest that the weakening of *cGAS*/STING responses to transfected DNA during cell division is associated with STING. Although binding to mitotic chromosomes reduces *cGAS* activity, it has no effect on Golgi integrity. Thus, the vesiculated state of the mitotic Golgi could serve as an additional protective measure to limit the potential harm caused by *cGAS*/STING responses to self-DNA [75]. RNAs, including transfer RNA (tRNA) and total RNA from cells, do not directly activate *cGAS* but instead promote its phase separation in vitro [76]. The formation of *cGAS* phase-separated biomolecular condensates in the presence of RNA is significant for initiating and regulating the innate immune response to cytosolic dsDNA [76]. This mechanism ensures a balanced immune response, preventing under or overactivation of *cGAS* [76].

Excessive *cGAS*/STING pathway activation, often due to genetic mutations, is linked to various harmful autoinflammatory and senescence responses. Mutations in STING or dNases TREX1 and DNASE2 can cause type-I interferonopathies, which prevent the elimination of DNA that triggers *cGAS* [77]. Additionally, mutations in STING are associated with conditions such as STING-associated vasculopathy, with onset in infancy (SAVI) and lupus-like syndromes [78].

The findings of phase-separated biomolecular condensates of *cGAS* not only increased our understanding of innate immunity’s physical foundations but also unveiled intricate methods for modulating cellular signaling. These insights have led to the development of novel therapeutic agents targeting the *cGAS*-STING pathway. These agents work through various mechanisms: dissolving *cGAS*-DNA phase-separated condensates (e.g., Oleic Acid) [79], chelating essential ions such as zinc (e.g., Zn-ejector TPEN) [59], or directly inhibiting the phase separation process (e.g., XQ2 and its analogs) [80].

The role of the Golgi apparatus in sustaining innate immunity signaling is complex. It is not only a conduit for the translocation of STING but also actively participates in its functional activation. This involvement of the Golgi underscores its significance in the broader context of intracellular signaling and immune response modulation. The relocation of STING from the ER to the Golgi, mediated by the interaction with *cGAMP*, is a critical step in the conversion of STING from its inactive to its active form. This step is fundamental in shaping the activation dynamics of STING and, by extension, the body’s ability to respond to bacterial infections effectively. The precise mechanisms by which the Golgi apparatus influences STING activation and the implications of this for immune signaling continue to be areas of active research and interest.

## 4. Golgi Apparatus and TLR/RLR Signaling

Viral infections have evolved strategies to exploit host factors for their replication within target cells, distinguishing them from bacteria, fungi, and helminths. The distinct molecular patterns present in viral genomes and replication intermediates render them discernible as pathogens, allowing the immune system to detect viral RNA while tolerating self-RNA. This delicate balance forms the basis of an effective antiviral response without inadvertently activating host systems. To achieve this, various families of RNA sensors have been identified. Among them, RIG-I-like receptors (RLRs) function in the cytosol to detect viral RNA, while Toll-like receptors (TLRs) recognize it within endolysosomes [81].

The Golgi apparatus, a central hub for cellular trafficking and processing, emerges as a pivotal player in the orchestration of these immune pathways. While the focus remains on RLR and TLR signaling, we will elucidate the integral role of the Golgi apparatus in enhancing these processes.

Upon recognizing pathogen-associated molecular pattern (PAMP) RNA, RLRs such as RIG-I and MDA5 undergo conformational changes, releasing their caspase activation and recruitment domains (CARDs) from a repressed state [82]. These CARDs then engage an adaptor molecule known as mitochondrial antiviral signaling protein (MAVS). Notably, the Golgi apparatus plays a significant role in these interactions, acting as a platform where MAVS forms prion-like aggregates [83]. These aggregates attract E3 ubiquitin ligases and downstream effector proteins like TNF receptor-associated factor 2 (TRAF2), TRAF3, and TRAF6, forming an active signaling complex termed the “signalosome”. This intricate cascade leads to phosphorylation, nuclear translocation of key innate immune transcription factors, including interferon regulatory factor 3 (IRF3) and IRF7, and the activation of NF-kB. Ultimately, this cascade results in the induction of type I and III interferons, as well as genes encoding proinflammatory cytokines and chemokines, collectively working to restrict viral infections [84].

The Golgi apparatus further comes into focus in the context of TBK1, a central component of this pathway (see Figure 2). TBK1′s critical role in type I interferon (IFN) production becomes evident, as its absence significantly diminishes the induction of type I IFNs, which are crucial for antiviral immune responses. Notably, some studies highlight that upon stimulation of RLR or TLR3 pathways, TBK1 can be targeted to the Golgi apparatus through its interaction with optineurin (OPTN), an adaptor protein binding to ubiquitin [83]. This interaction initiates the assembly of complexes between TBK1 and OPTN, leading to trans-autophosphorylation and activation of TBK1 [83]. Subsequently, TBK1 phosphorylates interferon regulatory factor 3 (IRF3) [85], a key step in promoting type I IFN production and triggering an effective antiviral response [86].

Furthermore, the Golgi apparatus’s impact extends to the regulation of interferon responses through TLR activation and intracellular cytoplasmic receptors like protein kinase R (PKR). The complex network of TNF receptor-associated factors (TRAFs) links upstream receptor signals with downstream gene activation. Of special note is TRAF3, which facilitates IRF3 activation through TNF receptor-associated factors, leading to IFN-beta (IFN-b) production [87].

As we delve into the role of the Golgi apparatus in these immune pathways, it becomes evident that the Golgi’s dynamic and multifaceted functions extend beyond protein trafficking.

Upon the identification of double-stranded RNA and double-stranded DNA, the Golgi apparatus exhibits a fragmentation process, resulting in the formation of punctate structures within the cytoplasm. These structures are instrumental in promoting the colocalization and subsequent interaction between TNF receptor-associated factor 3 and the mitochondrial antiviral-signaling protein. MAVS serves as an essential mitochondrial-associated adaptor molecule for the retinoic acid-inducible gene I-like helicases (RLHs). Investigations employing gene deletion techniques have underscored the pivotal role of TRAF3 in initiating the production of type I interferon through the RLH signaling pathway [88].

In the described signaling pathway, TNF receptor-associated factor 3 functions as a positive regulatory adaptor by interacting with mitochondrial antiviral-signaling protein and TNFRSF1A-associated via death domain (TRADD). This interaction is crucial for the initiation of IRF-3 phosphorylation, which involves the adaptor molecule TRAF family member-associated NF-kappa-B activator and the κB kinase (IKK)-related kinases TANK-binding kinase 1 and IKK-i [89,90]. Additionally, the recruitment of Fas-associated protein with death domain (FADD) and receptor-interacting protein 1 (RIP1) to MAVS by TRADD enhances the interaction between TANK and TRAF3. A proposed model suggests that TRADD orchestrates both FADD- and RIP1-mediated NF-κB signaling in conjunction with TRAF3- and TANK-mediated IRF-3 signaling [90,91]. However, further investigations are required to clarify the mechanisms underlying TRAF3′s recruitment to MAVS during viral infections.

Recent investigations have unveiled novel interactors of TRAF3, such as Sec16A and p115:Sec16A, identified as KIAA0310, and p115, identified as USO1, play a fundamental role in the endoplasmic reticulum-to-Golgi vesicular transport process. These proteins are primarily involved in facilitating anterograde trafficking at the interface between the ER and the Golgi apparatus, significantly influencing the assembly and transport of COPII vesicles. Sec16A, which assembles on the ER membrane, forms structural scaffolds that delineate the ER exit sites, essential for the assembly of COPII [92,93,94]. p115, characterized as a coiled-coil molecule with myosin-like properties, functions as a tethering adaptor. It is crucial for vesicle tethering at various locations, including the ER [95], the ER-Golgi Intermediate Compartment [88], and, in conjunction with giantin and GM130, at the cis-Golgi network.

Co-immunoprecipitation and confocal microscopy techniques have verified the association and colocalization of TNF receptor-associated factor 3 with p115 and Sec16A. Furthermore, experimental overexpression of p115 or Sec16A has been observed to augment the type I interferon response. Conversely, the reduction or suppression of these proteins impairs the induction of antiviral genes. Additionally, following the activation of the RIG-I-like helicases (RLH) and cytoplasmic DNA sensor pathways, there is a noted fragmentation of the Golgi apparatus into cytoplasmic punctate structures. This fragmentation process facilitates the colocalization and association of TRAF3 with the mitochondrial antiviral-signaling protein [93].

p115, identified as a vesicle-tethering protein, demonstrates specific colocalization and interaction with the amino-terminal region of the cis-Golgi protein GM130. Additionally, TRAF3, when tagged with the FLAG epitope (FLAG-TRAF3), also exhibits localization to the Golgi apparatus, displaying significant spatial overlap with the cis-Golgi marker GM130. This indicates that TRAF3 is present in the compartments involved in transport from the endoplasmic reticulum to the Golgi, where it is closely associated [96].

Exposure to double-stranded RNA mimic poly I:C or the dsDNA mimic poly dA:dT disrupts the typical ribbon-like structure of the Golgi apparatus, resulting in the formation of Golgi ministacks containing GM130. Importantly, the localization of endogenous TRAF3 parallels these fragmented Golgi structures. Comparable patterns were observed in cells infected with RIG-I inducers such as Sendai virus (SeV), Respiratory Syncytial Virus (RSV), and Influenza virus [97].

FLAG-tagged TRAF3 has been identified as localizing to the Golgi apparatus, exhibiting significant colocalization with the cis-Golgi marker GM130. Additionally, TRAF3 demonstrates a notable presence in ER-to-Golgi transport compartments.

New research has recently shown that the GTPase-trafficking protein RAB1B plays a crucial role in enhancing the signaling of the RIG-I pathway, which in turn promotes the induction of interferon-beta (IFN-β) and the overall antiviral response. In the context of viral infection, cellular proteins such as RAB1B undergo alterations in their localization, particularly to and from mitochondria-associated membranes. Following the activation of RIG-I, RAB1B is recruited into MAMs, where it interacts with the mitochondrial antiviral signaling protein signaling complex [98]. RAB1B, functionally distinct from its homolog RAB1A, modulates endoplasmic reticulum-to-Golgi trafficking. It achieves this by binding to effector proteins that facilitate the connection of ER-derived vesicles to the cytoskeleton, thereby aiding their transport to the Golgi [99].

In its active, GTP-bound, membrane-associated form, RAB1B is essential for facilitating the recruitment of coat protein complex I and the transport of cargo between the ER and the Golgi. This process is mediated through interactions with specific effector proteins. Furthermore, RAB1B plays a significant role in the signaling processes that lead to the autophosphorylation of TANK-binding kinase 1 and the subsequent phosphorylation of interferon regulatory factor 3 in response to RIG-I signaling [100].

As more and more adaptor proteins are found to interact with key players in the immune response signaling, the Golgi apparatus and the vesicular transport complexes that link it to the endoplasmic reticulum should be perceived not only as a platform for protein trafficking but dynamic structures that modulate downstream events of antiviral immunity signaling pathways.

## 5. Golgi Apparatus and Microbial Infection

Any alteration or fragmentation in the Golgi apparatus’s structural integrity can lead to a range of disorders, including infections [101]. Stress conditions like oxidative or pharmacological stress, cargo overloading, and ionic imbalance can induce “Golgi stress”, causing defects in membrane trafficking [102]. This delicate balance required for Golgi function underlines its vulnerability to microbial exploitation.

The “Golgi stress” concept encapsulates the organelle’s response to damage, involving autoregulation and signaling pathways like procaspase-2/golgin-160 and CREB3-ARF4 [103]. If the damage is beyond repair, the Golgi complex disintegrates, leading to cell death [103].

Golgi fragmentation, observed in various infectious diseases, serves dual purposes: helping infected cells escape the immune response and enhancing viral replication [101]. For example, during human rhinovirus infection, the Golgi is broken down into vesicles used as a membrane source for virus assembly [104].

Bacteria like Legionella pneumophila demonstrate remarkable sophistication in targeting the Golgi, rerouting ER-to-Golgi trafficking to facilitate replication within the Legionella-containing vacuole (LCV) [105]. This strategic manipulation involves a suite of effector proteins, highlighting the essential role of Golgi dynamics in bacterial pathogenesis. Similarly, *Chlamydia trachomatis* interacts closely with the Golgi, forming an inclusion body that hijacks lipid trafficking, emphasizing the Golgi’s role in the bacterial lifecycle [101]. Salmonella Typhimurium also exploits Golgi dynamics, positioning its replicative near the Golgi and manipulating Golgi-derived vesicles, which is crucial for its survival and pathogenesis [101].

Understanding the Golgi apparatus’s role in microbial infection highlights the potential for identifying new targets for disease diagnosis and therapy. Markers like Golgi glycoprotein 73 (GP73) [105,106,107] or anti-Golgi antibodies (AGAs) [48], elevated in various viral infections, could serve as valuable tools for therapeutic intervention. This intricate interplay between the Golgi apparatus and microbial pathogens emphasizes the critical need for further research to leverage this relationship in disease management and the development of new therapeutic strategies.

## 6. Conclusions

After many years of studying and debating its existence, the Golgi apparatus seems to function not only as a central organelle in vesicular trafficking and protein and lipid transport but also as a platform for innate immunity signaling and subsequent effectors. This review summarizes the new discoveries on how Golgi connects intracellular compartments such as ER and mitochondria and, therefore, plays a vital role in cell defense. Although many new studies are revealing the crucial part that this dynamic organelle plays in innate immunity, we need to better understand the detailed mechanisms that regulate the cellular response to pathogens. NLRP3 inflammasome formation still needs to be better investigated, and perhaps other roles of Golgi in innate immune pathways will be brought to light by further examining the ways that cellular compartments such as ER and mitochondria communicate with each other to orchestrate cellular defense. Additionally, the type I Interferon response initiated via the activation of Toll-like receptors is vital in limiting viral infections and, as many emerging studies show, is regulated at the Golgi by TBK1, situating Golgi in the first line of cell defense. By better understanding the signaling pathways that stand at the molecular base of innate immunity, we may be able to reveal how exactly we can intervene to potentiate the innate immune system to better fight microbial infection and refresh the way we view the immune response as a complex and very thoroughly regulated molecular process that happens due to the collaboration of many intracellular compartments in every infected cell. Finally, continuous research is much needed to reveal ideas and new therapeutical approaches in treating both infectious and inflammatory diseases and even improve the evolution of autoimmune diseases, a heavy burden for millions of patients around the world.

## Figures and Tables

**Figure 1 ijms-25-04120-f001:**
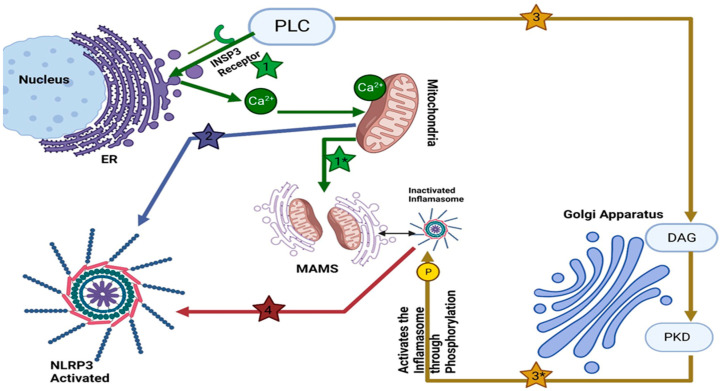
The figure shows PLC releases Ca^2+^ from the ER through inositol-1, 4, 5-trisphosphate (InsP3) Receptor. Ca^2+^ accumulates in the mitochondria, which may lead to mitochondrial damage (1). Damaged mitochondria release calcium ions (Ca^2+^) that accumulate in the mitochondria-associated membranes (MAMs), contributing to their activation (1*). Damaged mitochondria release several factors that trigger the activation of the NLRP3 inflammasome (2). NLRP3 was shown to directly bind to mitochondria-associated ER membranes (MAMs) (black arrow). The other product of PLC activation, diacylglycerol (DAG), accumulates in the Golgi (3) and increases the activity of PKD (protein kinase D), which then phosphorylates NLRP3-Inflammasome and activates it (3*), releasing it from MAMs (4).

**Figure 2 ijms-25-04120-f002:**
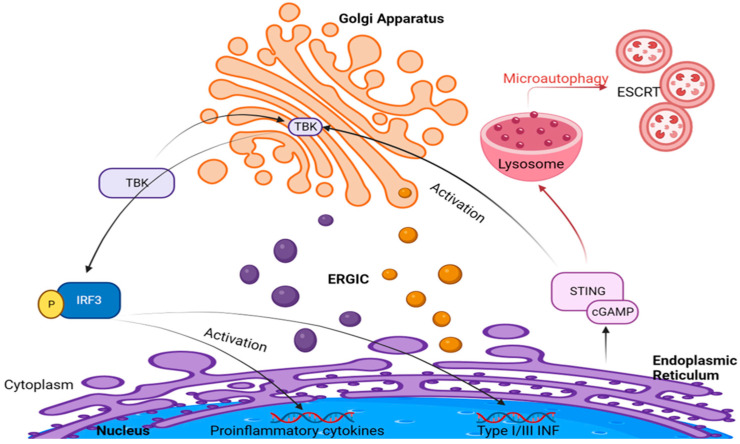
STING activates TBK1 at the Golgi. TBK1 phosphorylates interferon regulatory factor 3 (IRF3), leading to transcription of IFNb genes and stimulation of type I and III interferon genes as well as genes encoding proinflammatory cytokines and chemokines. STING is then degraded by the lysosomes through microautophagy.

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
