# Peer review of "The Golgi Apparatus: A Key Player in Innate Immunity"

_ijms, 2024, doi:10.3390/ijms25074120_

Round 1
Reviewer 1 Report
Comments and Suggestions for Authors
Dear Authors,
The manuscript is written well about the Golgi apparatus and its function in innate immunity, however some modification will be needed for publication in the journal. Reviewer's comments as below.
(1) Please follow submission guideline of this journal.
(2) Some citations in the manuscript were missing.
(3) Authors described "recent studies", but they cited papers that were published in 2018 or more old paper. Generally, recent paper should be published within 5 years.
Reviewer 2 Report
Comments and Suggestions for Authors
The manuscript entitled “The Golgi Apparatus: A Key Player in Innate Immunity” by MărunÅ£elu et al. is an excellent overview of the role the Golgi apparatus plays in the innate immune response. The review is structured in a very logical fashion and covers a several important areas of innate immunity and its relationship with the Golgi apparatus. The included figured are appropriate and help to demonstrate some of the more complex interactions described in the review.
Overall this is a very important and well written review on a topic that certainly needs to be considered when discussing innate immunity. Well done!
Reviewer 3 Report
Comments and Suggestions for Authors
Reviewer 4 Report
Comments and Suggestions for Authors
In this manuscript, the authors have provided a collection of information on the role of Golgi apparatus in innate immune response. This is an interesting review. The following points should be addressed before considering for publication.
1. A section on the innate immunity will improve the manuscript.
2. A separate section on how the Golgi apparatus regulate innate immunity after microbial infection should be included.
Comments on the Quality of English LanguageModerate English language editing will improve the manuscript.
Reviewer 5 Report
Comments and Suggestions for Authors
The Golgi apparatus is critical for protein processing and vesicular trafficking. This manuscript summarizes current understanding of the relationship between Golgi apparatus and innate immunity. This manuscript is well-organized; however, following points should be clarified.
Major points
#1: Authors have already published an excellent review for the Golgi apparatus in cells 2023 12, 1972. It may be nice to refer more deeply specific molecules that may associate with NLRP3, such as PKD (protein kinase D).
Round 2
Reviewer 3 Report
Comments and Suggestions for Authors
Authors have made adequate changes, additions and improvements.
Reviewer 4 Report
Comments and Suggestions for Authors
The manuscript was improved by revisions and can be accepted for publication.
Comments on the Quality of English LanguageModerate English language editing will improve the readability.